# Risk assessment of a highly pathogenic H5N1 influenza virus from mink

Katherine H. Restori[1,2], Kayla M. Septer[1,3], Cassandra J. Field[1,2,3], Devanshi R. Patel[1,3], David VanInsberghe [4,5], Vedhika Raghunathan [4,5], Anice C. Lowen [4,5] & Troy C. Sutton [1,2,3] ✉

Outbreaks of highly pathogenic H5N1 clade 2.3.4.4b viruses in farmed mink and seals combined with isolated human infections suggest these viruses pose a pandemic threat. To assess this threat, using the ferret model, we show an H5N1 isolate derived from mink transmits by direct contact to 75% of exposed ferrets and, in airborne transmission studies, the virus transmits to 37.5% of contacts. Sequence analyses show no mutations were associated with transmission. The H5N1 virus also has a low infectious dose and remains virulent at low doses. This isolate carries the adaptive mutation, PB2 T271A, and reversing this mutation reduces mortality and airborne transmission. This is the first report of a H5N1 clade 2.3.4.4b virus exhibiting direct contact and airborne transmissibility in ferrets. These data indicate heightened pandemic potential of the panzootic H5N1 viruses and emphasize the need for continued efforts to control outbreaks and monitor viral evolution.

Highly pathogenic H5N1 avian influenza viruses from the A/goose/Guangdong/1/1996 lineage pose a pandemic threat. Human infections with these viruses were first documented in China and Hong Kong in 1997[1–3]. Since then, there have been nearly 900 human cases worldwide with 50% mortality[4]. While these viruses emerged in Asia, they have been carried between continents by migratory birds and caused destructive poultry outbreaks in several countries. In 2021, a new H5N1 variant belonging to the subclade 2.3.4.4b rapidly dominated circulation in Asia, Africa, the Middle East, and Europe. Alarmingly, these viruses became endemic in wild-bird populations and expanded into avian species not typically affected. Their broad host and geographic ranges combined with high virulence led to ecological devastation, including large die-offs[5]. In late 2021, subclade 2.3.4.4b H5N1 viruses were first detected in North America. These viruses have now spread throughout North America and into South America where they continue to plague both wild and domestic bird populations. Across affected regions, subclade 2.3.4.4b H5N1 viruses have spilled over into several mammalian species with infections documented in skunks, foxes, bears, and seals[5,6]. Moreover, seven human infections with subclade 2.3.4.4b H5N1 viruses have been documented between January of 2022 and March 2023[7].

To cause a pandemic, an influenza A virus must be able to replicate efficiently in humans and transmit via the airborne route from person-to-person. Owing to similarities to humans in their susceptibility, ferrets are a valuable model in which to evaluate influenza virus transmission and pathogenesis, and ferrets are routinely used to assess pandemic risk. Ferrets possess a similar distribution of viral receptors (i.e.,α 2,6-linked sialic acids) to that observed in humans and, upon infection with human influenza viruses, ferrets develop clinical illness and shed high levels of infectious virus. Also consistent with influenza in humans, ferrets infected with human-adapted strains transmit the virus through the air to contact animals[8]. To date, no subclade 2.3.4.4b highly pathogenic H5N1 virus has exhibited the ability to transmit by the airborne route, a feature thought to be critical in limiting their outbreak potential in humans. However, experimental studies have demonstrated the potential for an ancestral clade 2.1.3.2 H5N1 virus to become airborne transmissible in ferrets[9].

[1]Department of Veterinary and Biomedical Science, The Pennsylvania State University, University Park, PA, USA. [2]Emory Center of Excellence of Influenza Research and Response (CEIRR), University Park, PA, USA. [3]The Huck Institutes of Life Sciences, The Pennsylvania State University, University Park, PA, USA. [4]Department of Microbiology and Immunology, Emory University School of Medicine, Atlanta, GA, USA. [5]Emory Center of Excellence of Influenza Research and Response (CEIRR), Atlanta, GA, USA. ✉e-mail: tcs38@psu.edu

In October of 2022, an outbreak of highly pathogenic subclade 2.3.4.4b H5N1 virus was reported in farmed mink in Spain[10]. Infected mink developed severe clinical disease that included reduced activity and feeding, bleeding from the snout, and neurological signs (ataxia and tremors). There was also evidence of mink-to-mink viral transmission; however, the mode of transmission could not be defined. To control the outbreak, all the mink on the farm were culled. During these efforts, farm workers were monitored for infection and no cases were detected among them. The use of face masks and other precautions implemented due to outbreaks of SARS-CoV-2 in mink may have offered protection[10]. Widespread infection of a mammalian species, and especially transmission among individuals, is concerning as it creates the opportunity for these avian influenza viruses to adapt to mammals, potentially increasing the risk of a pandemic.

To assess the risk to humans, we evaluated the potential for an H5N1 isolate from this mink outbreak to infect, cause disease, and transmit in ferrets. As isolates from the mink outbreak could not be readily obtained, we generated recombinant influenza A/mink/Spain/ 3691-8_22VIR10586-10/2022 (H5N1) virus [A/mink (H5N1)] using reverse genetics. Four whole genome sequences for viruses from the mink outbreak had previously been deposited in repositories (i.e., GISAID) and were publicly available. Two viruses shared the same amino acid changes across the genome compared to the closely related viruses of the A/gull/France/22P015977/2022-like genotype, while the remaining two viruses each had unique mutations[10]. Therefore, we selected one of the two viruses with shared amino acid changes for our

analysis. All viruses from this outbreak carried the mammalian-adaptive mutation T271A, and several additional mutations were identified throughout the genome; however, as previously reported, the function of these later mutations is unknown[10]. Virus rescue or regeneration, and all subsequent experiments were performed following strict biosafety protocols in our biosafety level 3 enhanced facility, and all studies were conducted following all local, state, and federal rules and regulations.

## Results

### Evaluation of contact and respiratory transmission

To evaluate direct contact transmission, donor ferrets were inoculated with $10^6$ tissue culture infectious dose 50% (TCID50) of the A/mink (H5N1) virus and were allowed to recover for 24 h. Subsequently, each infected donor (DR) ferret and an uninfected direct contact (DC) animal were co-housed in the same cage. To assess airborne transmission, DR ferrets were similarly inoculated and 24 h post-infection, each DR animal was paired with single respiratory contact (RC) animal in a transmission cage. The airborne transmission cage is designed such that the DR and RC ferrets cannot come into direct physical contact, but the animals share the same airspace. For both direct contact and respiratory transmission, nasal wash samples were collected from each animal on alternate days for 13 days to detect replicating virus. At the time of euthanasia, serum was collected and seroconversion against A/mink H5N1 was evaluated by hemagglutination inhibition (HI) assay.

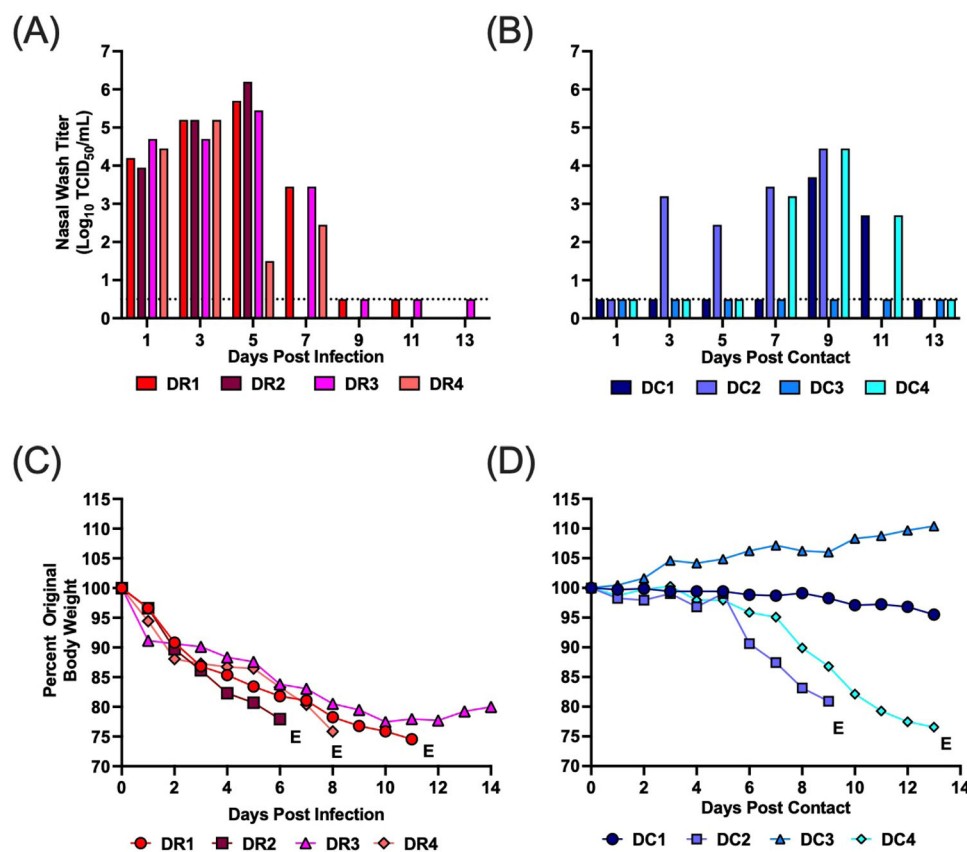

**Fig. 1 | Direct contact transmission, weight loss, and survival for ferrets infected with A/mink (H5N1).** Four donor ferrets were inoculated with A/mink (H5N1) and 24 h later each donor was paired with a contact in the same cage. Nasal wash samples were collected every other day for 13 days, and weight loss and clinical signs were monitored. **A**, **B** display viral titers from donor and direct contacts, respectively. **C**, **D** show weight loss from donors and direct contacts, respectively. DR and DC denote donors and direct contacts, respectively. Each bar or line represents an individual ferret. Males are DR1, DR2, DC1 and DC2, and females are DR3, DR4, DC3 and DC4. E denotes euthanasia. Ferrets DR1, DR2, and DR4, were euthanized on day 11, 6, and 8, respectively due to pronounced weight loss, diarrhea, and nasal discharge. DC2 and DC4 were euthanized on day 9 and 13, respectively, due to pronounced weight loss. Dotted line denotes limit of detection. Source data are provided as a Source Data file.

**Table 1 | Antibody titers in donor and contact ferrets for direct contact and airborne transmission studies with A/mink (H5N1) or A/mink (H5N1) PB2 A271T**

| Virus | Mode of transmission | Hemagglutination inhibition titers | | | |
|---|---|---|---|---|---|
| | | Donors | | Contacts | |
| A/mink (H5N1) | Direct contact transmission[a] | DR1 | 1:10[d] | DC1 | 1:80 |
| | | DR2 | 1:20 | DC2 | 1:20[d] |
| | | DR3 | 1:160 | DC3 | <1:10 |
| | | DR4 | 1:40 | DC4 | 1:20[d] |
| | Airborne transmission study 1[b] | DR1 | 1:160 | RC1 | 1:40 |
| | | DR2 | 1:160 | RC2 | n/p |
| | | DR3 | 1:20[d] | RC3 | <1:10 |
| | | DR4 | 1:320 | RC4 | <1:10 |
| | Airborne transmission study 2[c] | DR1 | n/p | RC1 | <1:10 |
| | | DR2 | 1:80 | RC2 | <1:10[d] |
| | | DR3 | 1:10 | RC3 | <1:10 |
| | | DR4 | 1:10 | RC4 | <1:10 |
| A/mink (H5N1) PB2 A271T | Airborne transmission | DR1 | 1:80 | RC1 | <1:10 |
| | | DR2 | 1:160 | RC2 | 1:20 |
| | | DR3 | 1:40 | RC3 | <1:10 |
| | | DR4 | n/p | RC4 | <1:10 |

Limit of detection = 1:10, n/p = not performed, animal had severe clinical disease and serum could not be obtained prior to euthanasia.

[a]Direct contact transmission: DR1 was euthanized on day 11, DR2 was euthanized on day 6, DR4 was euthanized on day 8. DC2 was euthanized on day 9 post-contact (p.c.), DC4 was euthanized on day 13 p.c.

[b]Transmission study 1: DR3 was euthanized on day 13, and RC2 was euthanized on day 12 p.c.

[c]Transmission study 2: DR1 was euthanized on day 7, DR3 was euthanized on day 9, DR4 was euthanized on day 8.

[d]Titers are low or animal did not seroconvert likely due to rapid progression to endpoint after onset of infection.

In the direct contact transmission studies, all donors became infected after inoculation and shed high titers of infectious virus in the nasal wash from days 1–7 post-infection (p.i.) (Fig. 1A). From days 1 to 12 p.i., all donor animals lost 20–25% body weight and three ferrets reached humane endpoint and were euthanized (Fig. 1C). Three of four DC ferrets showed evidence of replicating virus in their nasal wash with titers ranging from $10^{2.7}$–$10^{4.45}$ TCID50/mL (Fig. 1B). DC2 shed virus from days 3 to 9 post-contact (p.c.), DC4 shed virus on days 7 to 11 p.c., and DC1 shed virus on days 9, 11 p.c. (Fig. 1B). DC animals that shed virus also lost 3–25% body weight, and two contact ferrets reached humane endpoint and were euthanized (Fig. 1D). To verify transmission, hemagglutination inhibition assays (HI) were performed on serum collected at the time of euthanasia. All three contacts that shed virus also developed antibodies to A/mink (H5N1), although titers were low for the DC2 and DC4 animals likely due to reaching endpoint shortly after becoming infected. The one contact animal that did not shed virus (i.e., DC3) also did not seroconvert (Table 1).

As the H5N1 virus readily transmitted by direct contact, we proceeded to evaluate airborne transmission. We performed two separate studies using the same experimental design with four transmission pairs per study. In the first airborne transmission study, all donor (DR) ferrets became productively infected with high titers of infectious virus detected in nasal wash from days 1–7 p.i. (Fig. 2A). Between days 1 and 9 p.i. the donor animals lost 13–20% body weight (Fig. 2C), and one animal later reached humane endpoint criteria requiring euthanasia. In the paired respiratory contacts (RC), two of four animals showed evidence of replicating virus in the nasal wash (Fig. 2B). RC2 shed virus on days 9 and 11 p.c. with titers greater than $10^5$ TCID50/mL of nasal wash. This ferret also lost weight after day 8 and displayed neurological signs on day 12 p.c. requiring euthanasia (Fig. 2D). Replicating virus was also detected at low levels in nasal wash from the RC1 ferret on days 5 and

13 p.c. In compliance with our approved IACUC protocol nasal wash samples were not collected after day 13 p.c.; however, the RC1 animal lost weight starting on day 10 and exhibited 11.5% body weight loss on day 13 p.c. To confirm infection, we measured antibody titers against the H5N1 virus using serum collected on day 21 p.c. (Table 1). The RC1 animal developed antibodies against the A/mink (H5N1) virus, while the RC3 and RC4 animals did not seroconvert. Seroconversion was not assessed in RC2 as blood could not be collected (due to severe dehydration) prior to euthanasia. All 3 DR animals that survived until the end of the study also seroconverted (Table 1). The combined detection of replicating virus, clinical illness, and seroconversion in the RC1 and RC2 ferrets confirms these animals were infected by the airborne route. Therefore, the A/mink (H5N1) virus transmitted to 50% of respiratory contact animals in the first replicate experiment.

In the second study, consistent with the prior contact and airborne transmission studies, all donor ferrets became infected and shed high titers of virus from day 1–7 p.i. (Fig. 2E). All DR ferrets lost between 15–25% body weight by day 9 p.i., and three ferrets reached humane endpoint and had to be euthanized (Fig. 2G). One of four respiratory contact ferrets (i.e., RC2) shed virus at day 9, 11 and 13 p.c. (Fig. 2F). This ferret (RC2) lost 15% body weight and reached humane endpoint due to nasal hemorrhage requiring euthanasia (Fig. 2H). However, RC2 did not seroconvert (Table 1), most likely due to rapid progression to endpoint criteria with onset of shedding on day 9 p.c. and euthanasia on day 13 p.c. The measurement of replicating virus in the nasal wash, clinical illness and severe disease confirm that one ferret was infected by the airborne route in our second study. Thus, we report that A/mink (H5N1) transmitted to 25% of respiratory contact animals in the second study, and the combined transmission efficiency from both studies was 37.5%. Interestingly, for each transmission study we used two pairs of male and female ferrets and, of the three RC ferrets that became infected, all were males.

### Sequence analysis of A/mink (H5N1) viruses transmitted by direct and respiratory contact

To determine if adaptive mutations in the A/mink (H5N1) virus were associated with direct contact or airborne transmission, we performed a deep sequencing analyses of viruses shed in the nasal wash of infected donors and contacts. To enhance the detection of variants, viral RNA was isolated and sequenced from nasal wash samples on the day of peak titer for each animal. We identified more non-synonymous and synonymous single nucleotide variants in the infected direct contact ferrets compared to infected respiratory contacts. However, no synonymous or non-synonymous mutations were shared or consistently identified in either direct or respiratory contact animals, and no mutations consistently became enriched in the contact animals compared to donors (Fig. S1 and Table S1). Collectively, this analysis shows that direct contact and airborne transmission was not linked to changes in the A/mink (H5N1) virus.

### Assessment of virulence and infectious dose

To evaluate virulence of the A/mink (H5N1) virus, we performed a dose de-escalation study. This approach allows quantification of the median infectious dose and evaluation of disease severity across a range of doses likely to be representative of natural exposures. Groups of ferrets ($n = 4$/group, equal numbers of male and females/group) were inoculated with 10-fold decreasing doses of virus from $10^3$–$10^0$ TCID50. Nasal wash samples were then collected every other day for 9 days and clinical disease was monitored for 14 days. At inoculation doses of $10^1$–$10^3$ TCID50, all animals became infected, lost weight, and exhibited severe clinical disease consistent with that observed in farmed mink[10] (Fig. 3). Clinical signs included fever, reduced activity, respiratory symptoms, and diarrhea (see Tables S2 and S3 for detailed clinical scoring). Ferrets from groups infected with both low and moderate infectious doses exhibited increases in body temperature

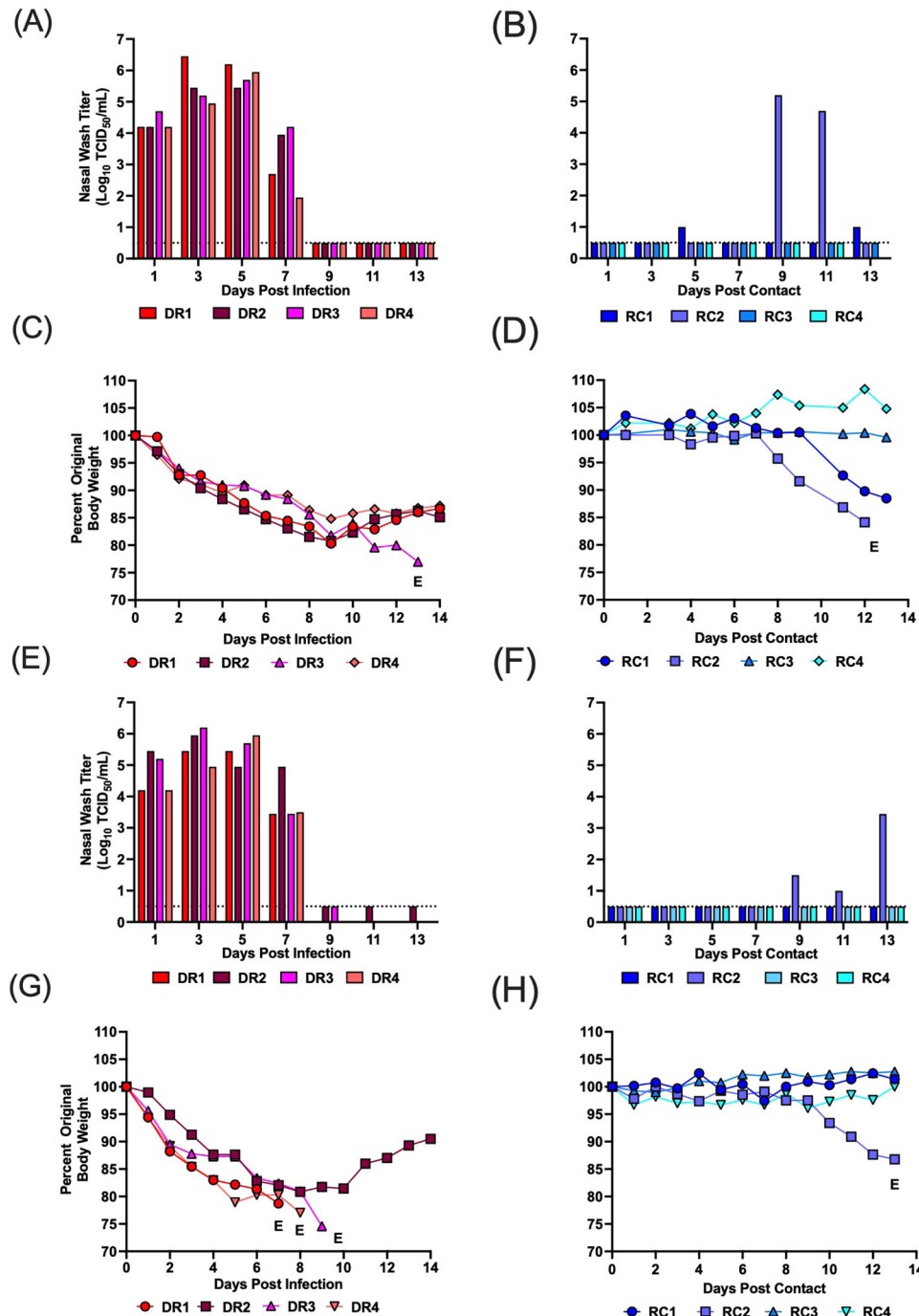

**Fig. 2 | Airborne transmission, weight loss and survival for ferrets infected with A/ mink (H5N1).** Two separate airborne transmission studies were performed. In each study, four donor ferrets were inoculated with A/mink (H5N1) and 24 h later each donor (DR) was paired with a single respiratory contact (RC) in a transmission cage. Nasal wash samples were collected every other day for 14 days, and weight loss and clinical signs were monitored. Transmission study 1 is depicted in (**A**–**D**), and transmission study 2 is shown in (**E**–**H**). Transmission study 1: **A**, **B** display viral titers from DR and RCs, respectively. **C**, **D** show weight loss from DR and RCs, respectively. Transmission study 2: **E**, **F** display viral titers from DR and RCs, respectively. **G**, **H** show weight loss from DR and RCs, respectively. Each bar or line represents an individual ferret. Males are DR1, DR2, RC1, and RC2 and females are DR3, DR4, RC3 and RC4. E denotes euthanasia. Study 1: ferret DR3 was euthanized on day 13 due to pronounced weight loss and RC2 was euthanized on day 12 post-contact due to neurological symptoms (hind-limb paralysis). Study 2: ferret DR1, DR3, DR4 were euthanized on days 7, 9, and 8, respectively, due to pronounced weight loss, and RC2 was euthanized on day 13 post-contact due to nasal hemorrhage. Dotted line denotes limit of detection. Source data are provided as a Source Data file.

indicative of fever (Table S2). One animal exhibited bleeding from the nose, while another animal had bleeding from the anus. Multiple animals also exhibited neurological disease including ataxia and hindlimb paralysis. Except for one animal in the group inoculated with $10^3$ TCID50, all the ferrets inoculated with virus doses of $10^1$–$10^3$ TCID50 reached humane endpoints and were euthanized (Fig. 3I–K). In the

group of animals infected with $10^0$ TCID50, one animal shed replicating virus in the nasal wash, lost weight, and reached humane endpoint (Fig. 3D, H, L). A second animal also lost weight and developed neurological signs requiring euthanasia but did not shed replicating virus in the nasal wash. This suggests the virus may have been replicating in the lungs and/or other extrapulmonary organs. The remaining two

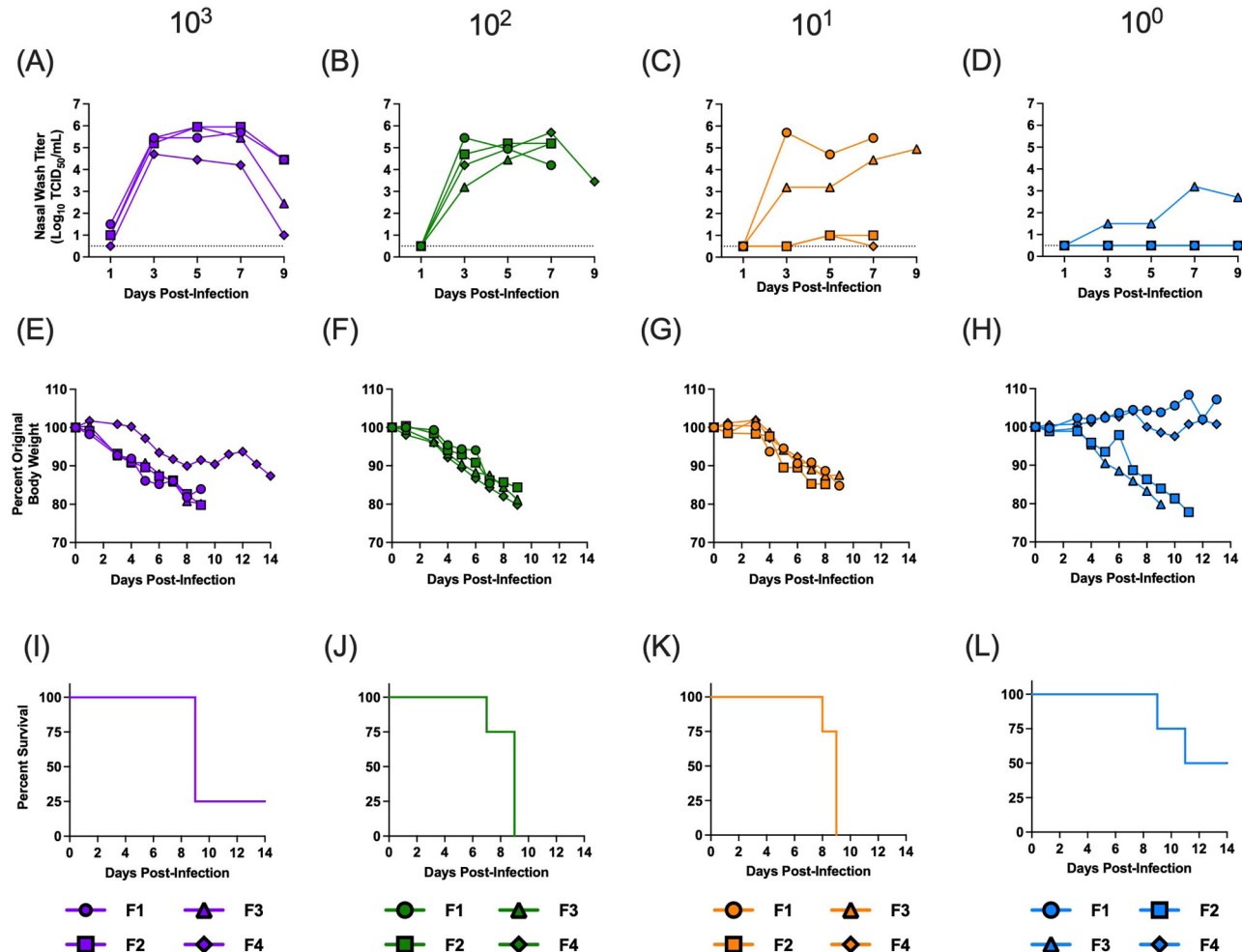

**Fig. 3 | A/mink (H5N1) virus has a low infectious dose in ferrets and remains virulent at low doses.** Groups of ferrets ($n = 4$/group) were inoculated with $10^3$ (purple), $10^2$ (green), $10^1$ (orange) or $10^0$ (blue) TCID50 of A/mink (H5N1). Nasal wash samples were collected every other day for 9 days and clinical signs were monitored for 14 days. The inoculation dose for each group of ferrets is indicated at the top of the figure. **A–D** Viral titers in the nasal wash samples at each inoculation dose, **E–H** weight loss in each ferret after virus inoculation, and **I–L** percent survival in each group of ferrets. For (**A–H**), each line represents an individual ferret. Males are F1 and F2 and females are F3 and F4. Nasal wash samples were titrated on MDCK cells and results are expressed as tissue culture infectious dose 50% (TCID50). Dotted line denotes limit of detection. Source data are provided as a Source Data file.

animals did not shed virus or show signs of disease. Importantly, we did not observe differences in mortality or symptom severity between male and females. Based on the criteria that viral shedding was indicative of infection, the 50% median infectious dose (ID50) was 3 TCID50 (Table 2), and if clinical illness was used to determine infection status, the ID50 was reduced to 1 TCID50. Strikingly, the A/mink (H5N1)

**Table 2 | Infectious dose 50% for A/mink (H5N1) and two pandemic influenza viruses**

| Inoculation dose | Proportion of infected ferrets | | |
|---|---|---|---|
| | A/Hong Kong/ 1/1968 (H3N2) | A/California/07/2009 (H1N1pdm09) | A/mink (H5N1) |
| $10^3$ | 4/4 | n/p | 4/4 |
| $10^2$ | 4/4 | 4/4 | 4/4 |
| $10^1$ | 3/4 | 4/4 | 4/4 |
| $10^0$ | 0/4 | 4/4 | 1/4 |
| Infectious dose 50% | 5 TCID50 | 1 TCID50 | 3 TCID50 |

When ID50 was determined to be a fractional unit of a TCID50, the ID50 was round up to the nearest whole infectious unit. Source data are provided as a Source Data file.

virus induced severe disease even when applied at a minimal infectious dose.

As median infectious dose is not commonly determined in ferrets, it was difficult to interpret the significance of a low infectious dose. To give a reference for comparison, we therefore determined the ID50 of representative human isolates from the 1968 H3N2 and the 2009 H1N1 influenza pandemics (A/Hong Kong/1/1968 (H3N2) [1968 H3N2] and A/California/07/2009 (H1N1pdm09) [2009 H1N1] viruses, respectively). Both viruses were generated using reverse genetics and groups of 4 ferrets were inoculated with 10-fold decreasing doses from $10^2$–$10^0$ TCID50. Neither pandemic virus caused severe disease in ferrets; however, the ID50 was very low. The 1968 H3N2 and 2009 H1N1 viruses had ID50 values of 5 TCID50 and 1 TCID50, respectively (Table 2). Thus, the ID50 for the A/mink (H5N1) virus is within the range of previous pandemic influenza viruses.

### Evaluation of the PB2 T271A mutation in virulence and airborne transmission

The A/mink (H5N1) virus possesses alanine (A) at position 271 of PB2 which differs from the avian consensus sequence PB2 271T for sub-clade 2.3.4.4b viruses isolated in Europe[10]. Prior studies have demonstrated that PB2 271A contributes to mammalian adaptation of avian

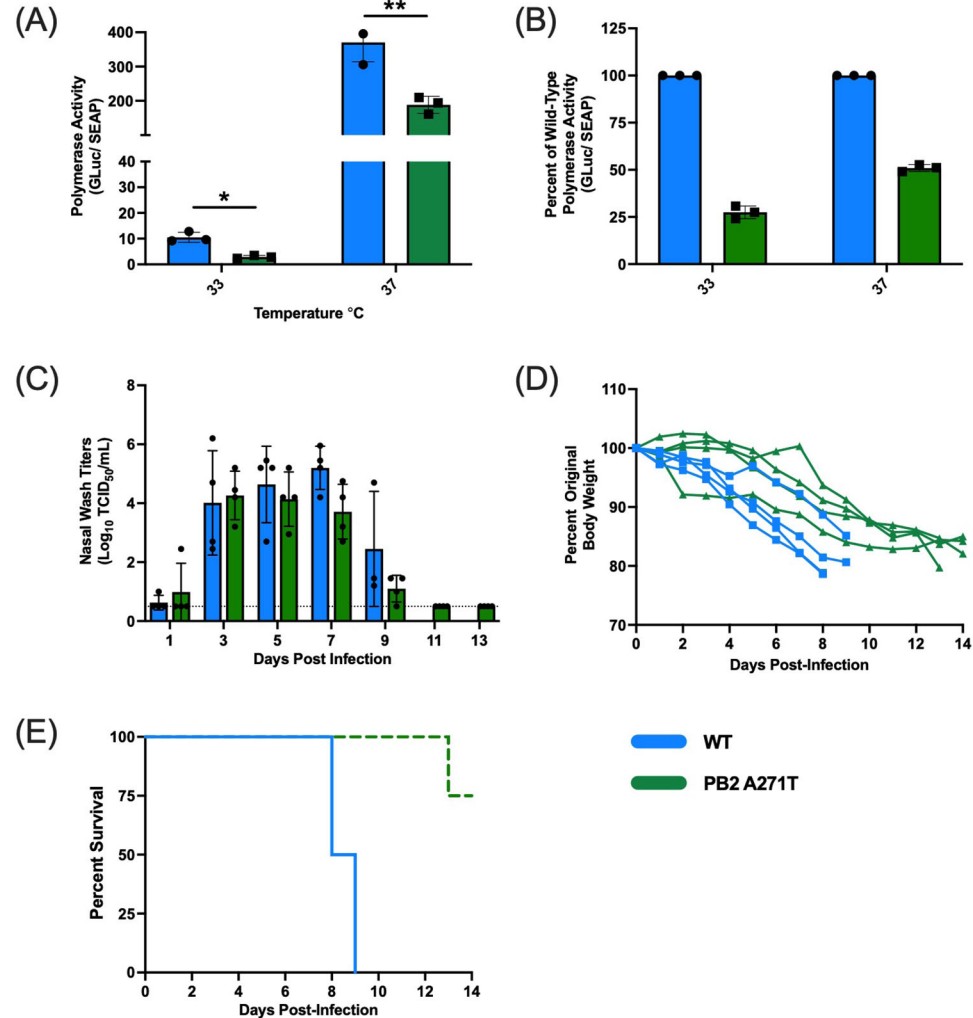

**Fig. 4 | PB2 A271T exhibits decreased polymerase activity in vitro and decreased mortality in ferrets relative to wild-type A/mink (H5N1).** Polymerase activity was determined by performing mini-genome assays in 293T cells at 33 and 37 °C. **A** Activity of the wild-type polymerase versus the polymerase with the mutation PB2 A271T at 24 h post-transfection. **B** Activity of the polymerase carrying the PB2 A271T mutation expressed as a percent of the activity of the wild-type polymerase. Data shown are mean ± SD from 1 of 3 representative independent experiments with $n=3$ biologically independent samples/experiment. **C–E** display ferret studies. Four ferrets per group were inoculated with $10^2$ TCID50/mL of either wild-type A/mink (H5N1) (i.e., PB2 271A) or A/mink (H5N1) with PB2 A271T mutation. Nasal wash samples were collected every other day, and weight loss and clinical signs were monitored daily for 14 days. **C** Displays viral titers as mean ± SD, **D** shows weight loss for individual animals, and **E** depicts survival ($n=4$/group). Blue and green bars and lines represent wild-type PB2 and PB2 A271T, respectively. Dotted line denotes limit of detection. *$p=0.015$ and **$p=0.018$ using a two-tailed unpaired Student's $t$ test with Welch's correction. Source data are provided as a Source Data file.

viruses[11–13]. Therefore, we next sought to assess the implications of this amino acid change for the phenotype of A/mink (H5N1). Using site directed mutagenesis we reversed the PB2 271A to 271T. To verify that this change altered polymerase function, we performed mini-genome assays. Introduction of PB2 A271T significantly reduced polymerase activity relative to the wild-type polymerase of the A/mink (H5N1) virus (Fig. 4A) at both 33 and 37 °C. When polymerase activity was expressed relative to wild-type, the PB2 A271T change reduced activity by 75% at 33 °C, and 50% at 37 °C (Fig. 4B).

Since we observed a decrease in polymerase activity when the PB2 mutation was reversed, we sought to assess the contribution of this mutation to virulence in ferrets. In the ID50 studies, an inoculation dose of $10^2$ TCID50 of recombinant wild-type A/mink (H5N1) consistently resulted in a productive infection with weight loss and 100% mortality (Fig. 3B, F, J). Therefore, to assess the role of PB2 A271T, groups of ferrets ($n=4$/group) were intranasally inoculated with wild-type A/mink (H5N1) or A/mink (H5N1) virus carrying the PB2 A271T mutation at a dose of $10^2$ TCID50. Nasal wash samples were then

collected every other day and weight loss, clinical signs, and survival were monitored. Consistent with our initial ID50 studies, all ferrets inoculated with the wild-type A/mink (H5N1) virus shed high titers of virus in their nasal wash and developed severe disease requiring euthanasia. Conversely, while all ferrets became infected with the PB2 A271T A/mink (H5N1) virus, mortality was reduced with only one of four ferrets reaching endpoint criteria requiring euthanasia (Fig. 4E). In parallel with reduced mortality, there were non-significant decreases in viral shedding from the nose, peak viral titers, and weight loss in the PB2 A271T group compared to wild-type group (Figs. 4C, D and 6A, C). Thus, introducing the PB2 A271T mutation in the A/mink (H5N1) virus reduced polymerase activity and mortality in the ferret model.

Next, we assessed airborne transmission of the A/mink (H5N1) virus with PB2 A271T mutation. Four donor ferrets were intranasally inoculated with $10^6$ TCID50 of this virus, and 24 h post-infection, these animals were paired with RC ferrets in our transmission cages. As observed with wild-type A/mink (H5N1) infection, all DR ferrets became infected and shed high titers of virus (Fig. 5A). However, only

(A)

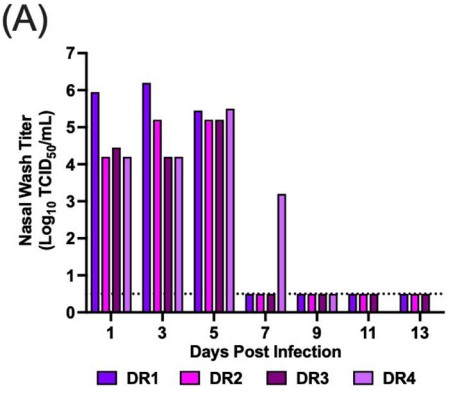

(B)

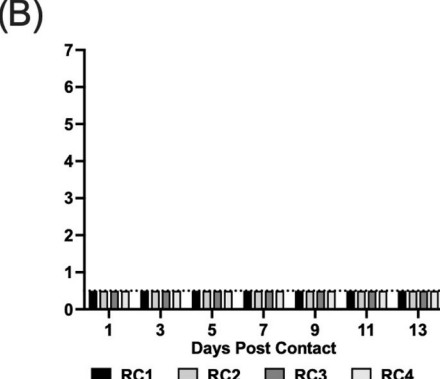

(C)

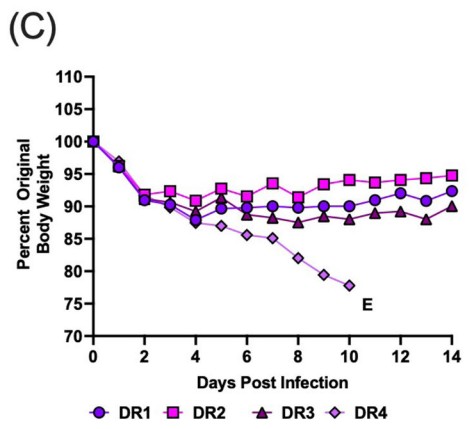

(D)

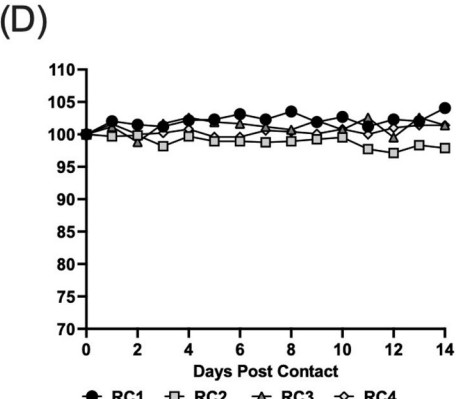

**Fig. 5 | Airborne transmission, weight loss and survival for ferrets infected with PB2 A271T A/mink (H5N1).** Four donor ferrets were inoculated with A/mink (H5N1) PB2 A271T and 24 h later each donor was paired with a single respiratory contact in a transmission cage. Nasal wash samples were collected every other day for 14 days, and weight loss and clinical signs were monitored. **A**, **C**, and **B**, **D** display viral titers and weight loss from individual donor and respiratory contacts, respectively. DR and RC denote donor and respiratory contact, respectively. Each bar or line represents an individual animal. Males are DR1, DR2, RC1 and RC2 and females are DR3, DR4, RC3 and RC4. E denotes euthanasia. Ferret DR4 was euthanized on day 10 due to pronounced weight loss. Dotted line denotes limit of detection. Source data are provided as a Source Data file.

one of four donor ferrets infected with PB2 A271T A/mink (H5N1) shed virus on day 7 p.i., while all DR ferrets infected with the wild-type virus shed high titers on day 7 p.i. (Fig. 2A, E). One of four DR ferrets infected with PB2 A271T A/mink (H5N1) virus lost 25% body weight and was euthanized in accordance with endpoint criteria (Fig. 5C). Interestingly, no RC ferrets shed virus or lost weight. Three RC animals did not seroconvert, while one animal (i.e., RC2) had an HI titer of 1:20. As this animal did not shed virus, the low HI titer suggests the animal was exposed to virus but did not develop a robust infection (Fig. 5B, D and Table 1). To determine if mutations emerged during viral replication of the PB2 A271T A/mink (H5N1) virus in DR ferrets, we sequenced viral RNA recovered from the nasal wash at the time of peak shedding in these animals. Consistent with transmission studies using the wild-type A/mink (H5N1) virus, we did not identify any synonymous or non-synonymous mutations that were shared by different donor ferrets, and no mutations were identified at a frequency above 8% (Fig. S1 and Table S1). Last, we compared the peak viral titers and total amount of virus shed from the DR ferrets infected with $10^6$ TCID50 of wild-type A/mink (H5N1) (Fig. 2, DR ferrets) or A/mink (H5N1) with PB2 A271T (Fig. 5, DR ferrets). We did not observe any significant differences in peak titers (Fig. 6B). However, animals infected with the A/mink (H5N1) PB2 A271T virus shed significantly less virus during the course of infection than ferrets infected with the wild-type virus as determined by area under the curve (Fig. 6D). Therefore, the PB2 A271T mutation led to a reduction in airborne transmission, as evidenced by a lack of

shedding in the RC ferrets, an outcome which may relate to an overall reduction in viral shedding from donor (DR) ferrets.

## Discussion

Collectively, our studies show that the A/mink (H5N1) virus transmitted efficiently by direct contact, with 75% of contact animals infected, and inefficiently via the airborne route with 37.5% of respiratory contact animals developing an infection. Sequence analyses of viruses shed in the nasal wash from infected donor and contact animals did not show evidence of positive selection acting during direct contact or airborne transmission. In dose de-escalation studies, the A/mink (H5N1) virus had a low infectious dose, and the virus was highly virulent across a wide range of doses as all infected animals developed severe disease. Although we observed efficient direct contact transmission, it was delayed relative to that seen previously with pandemic influenza viruses. For pandemic influenza viruses, viral shedding in direct contacts often begins on day 1 p.c. [14,15]. Here, the onset of shedding was on day 3 p.c. in one contact, and on days 7 and 9 p.c., respectively, in the others. While airborne transmission was observed, it occurred at lower efficiency than is typical for pandemic influenza viruses. Pandemic influenza viruses transmit to 75–100% of respiratory contact ferrets within 3–5 days after pairing [14,16–18]. By comparison, the A/mink (H5N1) virus transmitted to fewer contacts and with slower kinetics: in the first replicate, we first detected replicating virus in a contact on day 5 p.c., but clear evidence of productive viral replication did not occur until

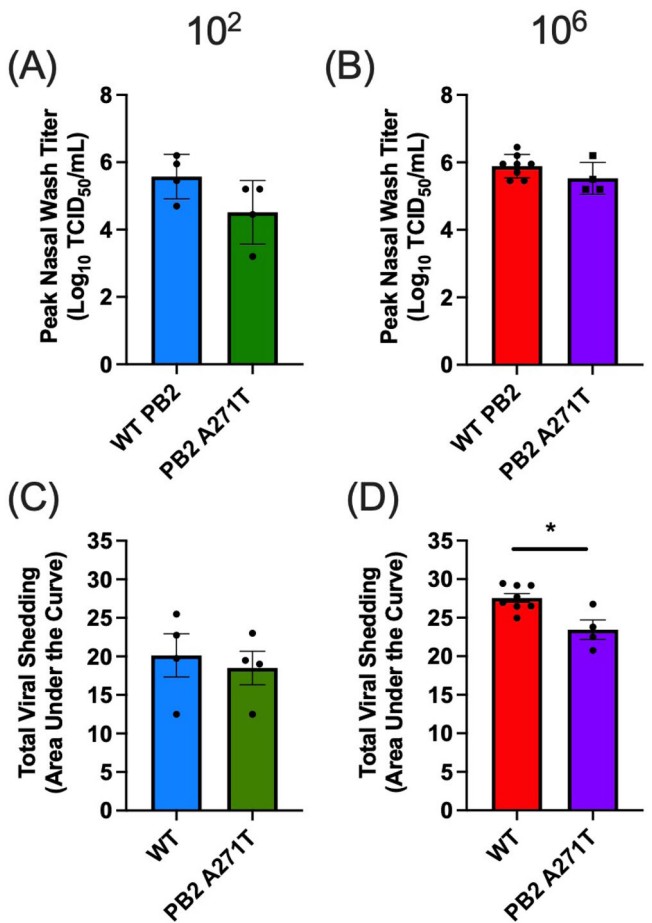

**Fig. 6 | Analyses of peak titers and total viral shedding for ferrets inoculated with wild-type or PB2 A271T A/mink (H5N1) virus. A–D** display peak viral titers and total viral shedding, respectively, for ferrets infected with either wild-type A/mink (H5N1) or A/mink (H5N1) with the PB2 A271T mutation. **A, C** show peak titers and total viral shedding, respectively, for ferrets infected at a dose of $10^2$ TCID50 ($n = 4$/virus) in studies comparing viral shedding, weight loss, and survival (Fig. 4). **B, D** compared donor ferrets from both airborne transmission study 1 and 2 (combined $n = 8$) that were inoculated with wild-type A/mink (H5N1) to the donor ferrets for the transmission study with PB2 A271T A/mink (H5N1) ($n = 4$) virus. For all transmission studies, the inoculation dose in donors was $10^6$ TCID50. Total viral shedding was determined via area under the curve (AUC) analyses of viral titers in the nasal wash from days 1–7. Values shown are mean ± SE. Peak titers and AUC values for each virus were compared using a two-tailed unpaired Student's *t* test with Welch's correction. *$p = 0.039$. Blue and green represent wild-type and PB2 A271T A/mink (H5N1), respectively, at an inoculation of $10^2$, while red and purple represent the same viruses, respectively, at an inoculation dose of $10^6$. Source data are provided as a Source Data file.

after day 9. In the second replicate, we detected replicating virus on day 9 p.c. and observed higher titers on day 13 p.c. Thus, the transmission observed in our studies is indicative of increased pandemic potential relative to previously characterized strains of highly pathogenic H5N1[9,19–25], but the A/mink (H5N1) virus does not exhibit transmission phenotypes consistent with pandemic strains.

Previous experimental work assessing the pandemic risk of H5N1 avian influenza viruses did not detect transmission of these viruses through the air. Unlike the A/mink H5N1 virus examined here, most of these studies were performed with isolates from birds or from clades older than 2.3.4.4b[9,19–25]. In one partial exception, 3 of 9 clade 2.2.1.2 highly pathogenic H5N1 viruses transmitted to respiratory contact ferrets in 1 or 2 of 2 transmission pairs; however, the transmissibility of these viruses could not be reproduced in subsequent experiments

using 4–6 pairs of ferrets[24]. A recent report also characterized recombinant subclade 2.3.4.4b H5N1 influenza viruses isolated from mink during the same outbreak as the mink isolate examined herein. In these studies, airborne transmission was evaluated using three pairs of ferrets for two isolates: A/mink/Spain/22VIR12774-13_3869-2/2022 (H5N1) and A/mink/Spain/22VIR12774-14_3869-3/2022 (H5N1). Neither isolate transmitted via the airborne route to contact ferrets[26]. As in our study, donor ferrets were inoculated with $10^6$ infectious units, and thus donor inoculation dose is unlikely to have contributed to differences in transmission. However, comparison of the amino acid sequences for these viruses relative to the A/mink/Spain/3691-8_22VIR10586-10/2022 (H5N1) strain used in our studies showed both isolates differed by 3 or more amino acids across the viral genome (Table S4). Changes were identified in the polymerase genes, neuraminidase, non-structural protein 1 and/or nuclear export protein (i.e., NS2), and it is possible these mutations may have contributed to differences in transmission.

Two additional risk assessment studies have evaluated the transmissibility of recent clade 2.3.4.4b H5N1 viruses isolated in North America[27,28]. In one, direct contact transmission was evaluated for two viruses: A/American wigeon/South Carolina/22-000345-001/2021 (H5N1) [Wigeon/SC/21] and A/Bald eagle/Florida/W22-134-OP/2022 [Eagle/FL/22][27]. The Wigeon/SC/21 virus is representative of early H5N1 viruses introduced into North America and has a similar genotype to viruses circulating in Eurasia. The Eagle/FL/22 virus carries PB2, PB1, and NP gene segments from North American low pathogenic avian influenza viruses, acquired by reassortment. In direct contact transmission experiments, none of the contact animals became infected or seroconverted; however, a striking difference in disease severity was observed in the donor ferrets. Donors infected with the Wigeon/SC/21 virus had mild disease, while all donors infected with the Eagle/FL/22 virus developed severe disease (i.e., weight loss, lethargy, and neurological signs) requiring euthanasia[27]. In a separate study, direct contact transmission of four North American lineage clade 2.3.4.4b viruses was evaluated[28]. In donor ferrets, the A/Red tailed hawk/Ontario/FAV-0473-4/2022 (H5N1) [RT.Hawk/ON/22] and A/Red fox/Ontario/FAV-301-05/2022 (H5N1) viruses caused severe disease and transmitted to 2/2 and 1/2 contacts, respectively. In follow-up studies with the RT.Hawk/ON/22 virus, donor ferrets were housed with both a direct contact and respiratory contact ferret. In this experimental system, 3 of 4 (75%) direct contact ferrets became infected and one respiratory contact had replicating virus recovered from a nasal swab on day 7 post-contact. On day 8 post-contact, all animals were euthanized to collect tissue samples, and replicating virus was not recovered in tissues from the airborne contacts[28]. During this transmission experiment, severe disease was again observed, with ferrets rapidly progressing to endpoint criteria. The observed 75% direct contact transmission is consistent with that seen here with the A/mink (H5N1) virus. Collectively, experimental data indicate that some clade 2.3.4.4b viruses of both the North American and Eurasian lineages transmit between ferrets that are in direct contact and that these viruses possess potential for limited airborne transmission. These findings emphasize the pandemic potential of clade 2.3.4.4b H5N1 viruses.

Importantly, we previously participated in an international exercise designed to evaluate the reproducibility of airborne influenza virus transmission in independent ferret model systems[29]. This exercise used a study design in which each donor ferret was paired with a single respiratory contact, and direct contact transmission was not assessed. We reported 100% airborne transmission of the 2009 pandemic H1N1 virus, consistent with 9 of 10 other groups, and we reported 25% airborne transmission of an avian H1N1 virus which was also observed by 2 of 9 other groups. One additional research group in this study reported 75% airborne transmission of the avian H1N1 virus. This study indicates that our experimental system is broadly comparable to other groups; however, it is important to highlight that our transmission studies are conducted in rooms with dedicated humidity

and temperature control systems that maintain environmental conditions within narrowly defined parameters. Throughout the experiments reported here, the temperature and relative humidity were maintained at $21 \pm 0.1\,°C$ and $36 \pm 4\%$, respectively. As prior studies in guinea pigs have shown these conditions favor airborne transmission of seasonal H3N2 and 2009 pandemic H1N1 viruses, and that increases in either temperature or humidity reduce transmission[30,31], our experimental system may have increased sensitivity for detecting airborne transmission.

The low median infectious dose observed for A/mink (H5N1) was comparable to the two most recent pandemic influenza viruses, indicating that the ability to infect and replicate in ferrets is not likely limiting transmission. Prior adaptation studies have shown that changes in the HA to reduce the pH of fusion and enhance binding to α2-6-linked sialic acids combined with mutations to enhance polymerase activity in mammalian cells are required for airborne transmission of a highly pathogenic clade 2.3.1.2 H5N1 virus[9,32]. The A/mink (H5N1) virus does not carry mutations known to reduce the pH of fusion or enhance binding to α2-6-linked sialic acids; however, the virus does carry the PB2 T271A mutation which has been shown to enhance polymerase activity and replication of avian viruses in mammalian cells. Consistent with this observation, we show that reversing the PB2 mutation (i.e., 271T) in A/mink (H5N1) reduced viral polymerase activity in mini-genome assays. Moreover, introducing the PB2 A271T mutation reduced mortality and resulted in a reduction in the number of RC ferrets that shed virus in airborne transmission studies. This was associated with reductions in viral titers in the nasal wash which were not significant at lower inoculation doses used to assess virulence but were significant at high inoculation doses used for transmission studies. These findings indicate the PB2 T271A mutation is enhancing viral replication of the A/mink (H5N1) virus contributing to both virulence and transmission in ferrets. As our studies assessed viral load in the nasal wash, future studies are warranted to assess the impact of the PB2 T271A mutation on viral replication in the lungs and systemic dissemination of the virus. Moreover, evaluating the role of the PB2 T271A mutation in the context of direct contact transmission will yield additional insight on the contribution of this mutation to the overall transmissibility of the virus.

An important consideration in interpreting our results with respect to the risk posed to humans is that the ferrets used in these studies have no pre-existing immunity to influenza, whereas the majority of humans have been exposed to H1N1 and H3N2 seasonal influenza viruses. While different influenza A virus subtypes are antigenically distinct, some degree of cross-protection against H5N1 may be conferred by prior exposure to these seasonal strains, especially against the N1 neuraminidase. Indeed, antibodies and T cells against seasonal influenza viruses have been shown to cross-react with H5N1 viruses[33–38]. Future studies are warranted to determine if prior immunity reduces disease severity and/or transmission. In conclusion, this is the first report of both direct contact and limited airborne transmission in a mammalian model of a subclade 2.3.4.4b H5N1 virus indicating these viruses pose a significant pandemic threat. Therefore, ongoing risk assessment and enhanced surveillance in wild and domestic animals is warranted to monitor the threat posed by these viruses as they continue to evolve and spillover into mammals, including humans.

## Methods

### Ethics statement, biocontainment, and animal care and use
All studies were conducted in compliance with all local, state, and federal rules and regulations. Experiments using the A/mink (H5N1) virus or derivatives were conducted in our Centers for Disease Control (CDC) and United States Department of Agriculture (USDA) approved high containment biosafety level 3 enhanced laboratory. All experiments were conducted in accordance with Institutional Biosafety

Committee Protocol No. 48971 and Institutional Animal Care and Use (IACUC) Protocol No. 201800250.

### Cell lines
Madin-Darby Canine Kidney (MDCK) Cells, London Line, FR-58, was obtained through the International Reagent Resource, Influenza Division, WHO Collaborating Center for Surveillance, Epidemiology and Control of Influenza, Centers for Disease Control and Prevention, Atlanta, GA, USA. Human embryonic kidney cells (HEK 293T) (CRL-3216) were obtained from the American Type Culture Collection. MDCK cells were cultured in DMEM (Cytiva) supplemented with 10% fetal bovine serum (Seradigm), 25 mM HEPES buffer, 2 mM L-glutamine (Corning) and 1% antibiotic-antimycotic (Gibco). 293T cells were cultured in the same media without HEPES buffer. All cells were cultured in a humidified incubator at $37\,°C$ with 5% $CO_2$.

### Generation, culture, and titration of viruses
Gene sequences for A/mink/Spain/3691-8_22VIR10586-10/2022 (H5N1) (GISAID Accession# EPI2220590-EPI2220597) were synthesized by Twist Biosciences and cloned into bi-directional reverse genetics plasmids[39]. For gene sequences missing the complete untranslated region, the missing region was replaced by that from A/mink/Spain/22VIR12774_14_3869-3/2022 (H5N1) (GISAID Accession # EPI2291511–EPI2291518). Reverse genetics plasmids were amplified in stable competent high efficiency *E. coli* cells (New England BioLabs), purified using a HiSpeed Plasmid Maxi-Prep Kit (Qiagen), and sequenced by Plasmidsaurus (Eugene, OR). Site-directed mutagenesis (QuikChange II XL, Agilent) was performed to generate the PB2 A271T mutant. Primers were as follows: Forward g811a_a813g_aaatatcgttaggag agcaacggtatcagcagacccattgg, Reverse a813g_ g811a_ccaatgggtctgctgata ccgttgctctcctaacgatattt (Integrated DNA Technologies). The virus was rescued by transfecting a MDCK-293T co-culture with all 8 reverse genetics plasmids using TransIT-LT1 (Mirus)[40]. Subsequently, to generate a virus stock, the virus was passaged twice in MDCK cells cultured in Opti-MEM media (Invitrogen) supplemented with 1 ug/mL of tosyl-sulfonyl phenylalanyl chloromethyl ketone (TPCK)-trypsin (Worthington). The TCID50 of the stock was determined by infecting 24-well plates of MDCK cells with serial dilutions of virus. Plates were incubated for 4 days at $37\,°C$ with 5% $CO_2$ and scored for cytopathic effect. The TCID50 was then calculated using the method of Reed and Muench[41]. Recombinant A/California/07/2009 (H1N1pdm09) [2009 H1N1] and A/Hong Kong/1/1968 (H3N2) [1968 H3N2] viruses were generated using the same methodology. Viral stock titers for A/mink (H5N1), A/mink (H5N1) PB2 A271T, 2009 H1N1, and 1968 H3N2 were $10^{7.5625}$, $10^{8.8125}$, $10^{7.125}$, and $10^{6.875}$ TCID50/mL, respectively.

### Ferret transmission experiments
Equal numbers of male and female ferrets 23 weeks of age were purchased from Triple F Farms (Gillett, PA). Ferret serum was screened by hemagglutination inhibition assay and animals were determined to be seronegative for A/Indiana/02/2020 (H1N1) and A/Tasmania/503/2020 (H3N2) viruses. For the direct contact experiment, ferrets were cohoused in the same cage. Airborne transmission experiments were performed using custom-built ferret transmission cages (Allentown). The cages are designed such that within each transmission chamber an infected donor (DR) ferret is separated from a respiratory contact (RC) ferret by two perforated offset stainless-steel plates with unidirectional airflow from the DR to RC animal. The cages are designed such that the animals cannot physically contact each other, but they share the same airspace. DR ferrets ($n = 4$, 2 male and 2 female) were sedated with a mixture of ketamine (20 mg/kg), xylazine (2 mg/kg), and atropine (0.05 mg/kg) and intranasally inoculated with $1 \times 10^6$ TCID$_{50}$ of recombinant A/mink/Spain/3691-8_22VIR10586-10/2022 (H5N1) or PB2 A271T r.A/mink/Spain/3691-8_22VIR10586-10/2022 (H5N1) in a 1-mL volume. After inoculation, sedation was reversed by administration of

atipamezole (1 mg/kg). Twenty-four hours later, DR ferrets were sedated, a nasal wash sample was collected using 1-mL of PBS. After the animal recovered, for direct contact studies, an uninfected sex-matched contact ferret was then co-housed with the infected donor, while for airborne transmission studies, a respiratory contact (RC) animal (also sex matched to the DR ferret) was housed adjacently to the infected donor. Nasal wash samples were then collected from the donor and contact ferrets on alternating days for 14 days as previously described[18,42].

To assess disease severity, ferrets were weighed and scored for clinical signs daily as described by Reuman et al.[43]. Animals were scored for activity, respiratory disease, weight loss, and other symptoms (e.g., diarrhea or neurological symptoms where a score of 0 denotes no signs and a score of 3 denotes severe symptoms). Ferrets were euthanized if weight loss exceeded 25% of day 0 body weight or 20% body weight loss and observation of another symptom above a score of 2 such as low activity, diarrhea with fecal material embedded in fur around the anus or opaque nasal/ocular discharge. Ferrets were euthanized immediately if neurological symptoms were present such as ataxia or hindlimb paralysis.

### Infectious dose studies
Infectious dose 50 (ID50) experiments were conducted by intranasally inoculating equal numbers of sedated male and female ferrets with $1 \times 10^3$, $1 \times 10^2$, $1 \times 10^1$ or $1 \times 10^0$ TCID$_{50}$ of A/mink (H5N1) in a 1-mL volume ($n = 4$/dose). Ferrets were sedated and nasal washed every other day for 9 days. Ferrets were monitored for clinical signs as described above and the ID50 was determined by the method of Reed and Muench[41].

### Evaluation of virulence of the PB2 A271T mutation
Four ferrets ($n = 4$, 2 male, 2 female) per group were inoculated with $10^2$ TCID50/mL wild-type A/mink (H5N1) or PB2 A271T A/mink (H5N1) in a 1 mL volume. Ferrets were sedated and nasal wash samples were collected every other day for 14 days. Weight loss and clinical symptoms were monitored daily.

### Titration of nasal wash samples
Nasal wash samples were titrated on MDCK cells using a combination of 24-well and 96-well plates to determine the TCID50/mL of nasal wash. To achieve a low limit of detection, 100 uL of nasal wash sample was incubated on 2 wells of a 24-well plate for 1 h. The nasal wash sample was then removed and replaced with virus culture media consisting of Opti-MEM media (Invitrogen) supplemented with 1 ug/mL of tosylsulfonyl phenylalanyl chloromethyl ketone (TPCK)-trypsin (Worthington) and 1% antibiotic-antimycotic (Gibco). To quantify peak titers in the nasal wash samples, 10-fold serial dilutions of virus were prepared in virus culture media and added to MDCK cells grown in 96-well plates (4 wells per dilution). Plates were then incubated in humidified incubator at 37 °C with 5% $CO_2$ and scored 96 h later for the presence of cytopathic effect (CPE). The TCID50/mL was then calculated using the method of Reed and Muench[42,44].

### Hemagglutination inhibition (HI)
Serum from donor and contact ferrets was treated with receptor-destroying enzyme (RDE)(Denken Seiken, Tokyo, Japan) overnight at 37 °C and then heat-inactivated at 56 °C for 30 min. Serum (25 uL) was then diluted 2-fold in a V-bottom plate (Corning) and mixed with 4 hemagglutination assay units (HAU) of A/mink (H5N1) virus prepared in 25 uL of PBS. After incubation at room temperature for 15 min, 50 uL of turkey red blood cells (0.5% solution in normal saline) (Lampire Biological Laboratories) was added to each well. The plate was then incubated for an additional 45 min and scored for hemagglutination inhibition[45].

### Mini-genome assay
Mini-genome assays were performed using human embryonic kidney 293T (HEK 293T) cells cultured in 6-well plates. Cells were transfected with reverse genetics plasmids (1 ug/plasmid mixed with TransIT-LT1 (Mirus)) encoding NP, PA, PB1, and PB2 or PB2 271T, of A/mink (H5N1) to assess the polymerase activity of wild-type PB2 in comparison to the polymerase carrying PB2 A271T. Cells were also transfected with Gaussia Luciferase (GLuc) and Secreted Alkaline Phosphatase (SEAP) reporter plasmids (generously provided by Dr. Daniel Perez, University of Georgia, Athens, GA) and then incubated at 33 or 37 °C[46]. Supernatants were collected at 24 h post-transfection, and luciferase and SEAP activity were assayed using a Secrete-Pair Dual Luminescence Assay Kit (GeneCopoeia, Inc.) and a Spec-traMax iD3 plate reader (Molecular Devices). Polymerase activity was expressed as a ratio of luminescence to SEAP activity for each sample. Three independent experiments were performed in triplicate and results were analyzed using an unpaired, two-tailed Student's $t$ test.

### Analysis of viral genome sequences
RNA was isolated from nasal wash samples collected from donors and respiratory contacts on the day of peak viral shedding using a RNeasy Mini Kit (Qiagen). vRNA was converted to cDNA using the M-RTPCR protocol[47]. Briefly, one-step reverse transcription PCR amplification of full viral genomes was achieved using pooled Uni12/Inf-1 and Uni12/Inf-3 forward primers and Uni13/Inf-1 reverse primers and SuperScript III Platinum kit (Thermo Fisher). Following PCR purification (Agencourt AMPure XP, Beckman Coulter) and determining concentration (Qubit dsDNA Quantitation High Sensitivity Assay, Thermo Fisher), cDNA was processed at the Emory National Primate Research Center (ENPRC) Genomics Core for sequencing on an Illumina NovaSeq 6000 platform. Samples were sequenced as $2 \times 100$ bp paired reads and demultiplexed prior to delivery.

Using the FluSAP pipeline developed in the Lowen laboratory (https://github.com/Lowen-Lab/FluSAP), reads were merged and filtered for low average quality (≥30) using BBMerge, then separated according to the segment using BLAT[48]. The reads were then mapped to their corresponding reference segment using BBMap, with local alignment set to false. From these alignments, we used Python scripts to identify iSNVs. Cutoffs for inclusion of iSNVs were set empirically. First, sites were evaluated based on their total coverage and the average quality and mapping statistics. Only sites with ≥100x coverage were considered. For minor variants at these sites to be included in subsequent analyses, they were required to be present at ≥1% frequency and have an average phred score of ≥35, and the reads that contained the minor allele at any given site also had to have sufficient mapping quality to justify inclusion. Specifically, reads containing the minor allele needed an average mapping quality score of ≥40, the average location of the minor allele needed to be ≥20 bases from the nearest end of a read, and the reads overall needed to have ≤2.0 average mismatch and indel counts relative to the reference sequence.

### Data and statistical analyses
Total viral shedding was determined by performing area under the curve (AUC) analyses using Prism (GraphPad v10). AUC analysis was restricted to days 1–7 post-infection as no animals reached endpoint criteria during this time. This ensured comparable periods of viral shedding were analyzed. For comparison of polymerase activity (Luciferase/SEAP), peak nasal wash titers, and AUC, values were confirmed to have normal distribution by Shapiro–Wilk test and were then compared using a two-tailed unpaired Student's $t$ test with Welch's correction. Statistical analyses were performed in Prism (GraphPad v10) with $p < 0.05$ considered significant.

**Reporting summary**

Further information on research design is available in the Nature Portfolio Reporting Summary linked to this article.

## Data availability

All data supporting the findings of this study are contained within the manuscript, in the Supplementary files, and/or Source data file. All raw sequencing data are available in NCBI's Sequence Read Archive under BioProject accession number PRJNA1098700. See www.ncbi.nlm.nih. gov/bioproject/PRJNA1098700. Source data are provided with this paper.

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

## Acknowledgements

We would like to acknowledge investigators at the Laboratorio Central de Veterinaria, Madrid, Spain, and Giacomo Barbierato at the Istituto Zooprofilattico Sperimentale delle Venezie, Padua, Italy, for contributing sequence information for the A/mink (H5N1) virus to the Global Initiative on Sharing All Influenza Data (GISAID) database. We would like to acknowledge the Pennsylvania State University Animal Resource Program and the BSL3+ Laboratory Managers for assistance with animal studies. This project has been funded in whole or in part with Federal funds from the National Institute of Allergy and Infectious Diseases, National Institutes of Health, Department of Health and Human Services, under Contract No. 75N93021C00017 (NIAID Centers of Excellence for Influenza Research and Response, CEIRR) (T.C.S. and A.C.L.). Support was also provided by the USDA National Institute of Food and Agriculture, Hatch project 4955 (T.C.S.).

## Author contributions

K.H.R., K.M.S., C.J.F., and D.R.P. conducted experiments with assistance from T.C.S. V.R. performed next generation sequencing. D.V. performed bioinformatic analyses. K.H.R., A.C.L., and T.C.S. drafted and revised the manuscript. T.C.S. conceived of the study, obtained research funding, and ensured regulatory compliance.

## Competing interests

The authors declare no competing interests.
