## [Peer Review File · Nature Communications]

Risk assessment of a highly pathogenic H5N1 influenza virus from minkEditorial Note: This manuscript has been previously reviewed at another journal that is not operating a transparent peer review scheme. This document only contains reviewer comments and rebuttal letters for versions considered at *Nature Communications*.

REVIEWER COMMENTS

Reviewer #1 (Remarks to the Author):

Authors have made substantial improvements to this manuscript from initial submission, primarily by repeating the airborne transmission experiment and including a second direct contact transmission experiment, including sequencing of nasal wash specimens, and clarifying wording throughout the text. Additional experiments included in this revision investigating the role an adaptation marker in PB2 contributes to virulence and transmission have further resulted in a study which is overall more comprehensive and complete.

Comments:

-Figure 6: Why did the authors not just report all n=8 wild-type ferrets inoculated with 10^6 TCID50 (panels B and C) together in one graph and compare to the n=4 mutant virus, and not (as it appears) re-copy the mutant virus data across panels B and C both showing results from n=4 from each study? Additionally, for area under the curve analyses, which day span of NW collections were included in this analysis, and how did authors account for animals reaching terminal endpoints prior to full collection of all NW specimens (75% of ferrets inoculated with 10^2 TCID50, and 25% of ferrets inoculated with 10^6 TCID50, were euthanized prior to collection of NW specimens that were negative for influenza virus)? Additional specificity in the methods regarding how this analysis was conducted and how authors controlled for situations where animals were euthanized during sample collection would be helpful (especially as the datapoints for the PB2 A271T area under the curve measurements in panels E and F appear to differ despite what appears to be identical datasets for this virus in panels B and C).

-because the serology data is helpful to contextualize the three transmission experiments, and the serology tables are all very small, consider including/embedding Tables 1, 2, and 4 into the same display images as Figures 1, 2, and 5 so the serology data is presented alongside the viral titer information (or combining all serology results into one stand-alone table).

-footnotes in Tables 1 and 2 indicating a lack of expected serological titers for specific animals are better described in the text itself as these assumptions do not reflect actual quantifiable data.

Reviewer #3 (Remarks to the Author):

Comments on "Risk Assessment of a highly pathogenic H5N1 influenza virus from mink"
(NCOMMS-24-09407-T)

The authors evaluated the pathogenicity and transmission potential of a clade 2.3.4.4b A(H5N1) virus [A/mink (H5N1)], which caused an outbreak in a mink farm in 2022. The virus was re-constructed by synthesized sequences using reverse-genetics and was tested using the ferret model commonly used for influenza risk assessment studies. The A/mink (H5N1) was transmitted from 3 out of 4 pairs of the inoculated donors to co-housed direct contact (DC) ferrets (75%) and from 3 out of 8 pairs of the inoculated donors to respiratory contact (RC) ferrets (37.5%). Under serial dilution, the ferret infectious doses was determined as 3 TCID50 (based on viral shedding

from inoculated animals) and 1 TCID₅₀ (based on clinical signs), which were comparable to that of the A/HK/1/1968 A(H3N2) and A/CA/07/2009 A(H1N1)pdm09 viruses. The A/mink (H5N1) virus possess an adaptive change, PB2-T271A, that has been shown to increase viral polymerase activity in mammalian cells. To evaluate the role of the PB2-T271A change on viral pathogenicity and transmissibility, the authors generated a recombinant virus carrying PB2-271T that is consensus among avian influenza viruses. The A/mink (H5N1) PB2-A271T virus was transmitted to 1 out of 4 RC (25%) and decreased lethality (from 100% to 25%) in ferrets under the inoculation dose of 100 TCID₅₀. Comparable peak viral loads were detected from ferrets inoculated with A/mink (H5N1) and A/mink (H5N1) PB2-A271T; however, the total amount of virus shed (area under the curve) by the A/mink (H5N1) inoculated ferrets were higher. Based on the results, the authors concluded that the clade 2.3.4.4b virus possess heightened pandemic potential relative to ancestral A(H5N1) strains.

The results have contributed to information to assess the pandemic risk of clade 2.3.4.4b H5N1 viruses, which share genetically similar HA gene but are otherwise genetically diverse in other gene segments. To date, several publications have reported risk assessment study of clade 2.3.4.4b in ferrets. Kendell et al. (PMID: 37248261) characterized two clade 2.3.4.4b H5N1 viruses isolated from USA and reported no transmission to DC ferrets. Further characterization of the genetically diverse clade 2.3.4.4b viruses observed different in pathogenicity in ferrets. Kobasa et al. (<https://www.researchsquare.com/article/rs-2842567/v1>) have also evaluated the transmission potential of clade 2.3.4.4b H5N1 viruses isolated from Canada, they reported that one virus was able to transmit efficiently to DC but not to RC ferrets. As the H5N1 outbreak in the mink farm has raised concerns for viral transmission potential in mammalian species, Maemura et al. (PMID: 37812908) have similarly constructed 2 recombinant viruses isolated from the same mink farm outbreak and tested their transmission potential in ferrets. They reported no reported no transmission of both viruses to RC ferrets. Taken together, the high genetic diversity of the clade 2.3.4.4b viruses may affect viral transmissibility and pathogenicity in ferrets. As such, the conclusion made by the authors that "clade 2.3.4.4b virus possess heightened pandemic potential relative to ancestral A(H5N1) strains" should be modified based on published literatures.

Specific comments

1. Please discussed other published risk assessment study of clade 2.3.4.4b viruses in ferrets.
2. H5N1 viruses have been reported to cause systemic spread in mammalian species and are generally replicating better in the lungs than in the nasal cavity. As such, viral load detected in the nasal washes may not fully reflect viral replication potential in ferrets. To confirm the effect of PB2-T271A change on viral pathogenicity in ferrets, the authors should compare viral titers in the lungs and other organs from ferrets inoculated with A/mink (H5N1) vs. A/mink (H5N1) PB2-A271T.
3. In regards of transmissibility, A/mink (H5N1) vs. A/mink (H5N1) PB2-A271T showed 25% and 37.5% transmissibility to RC ferrets, which is not statistically significantly different. Can the authors determine the DC transmission potential of A/mink (H5N1) PB2-A271T?
4. Line 296. The authors mentioned a study that compared the reproducibility of influenza transmission experiments. Can the authors provide a reference to this study?
5. Figure 6B,C,D,F, since the animals were inoculated with the same dose, would it be possible to combine the results to get a larger animal sample size, which may help to improve statistical analysis?

Response to Reviewers: Nature Communications manuscript NCOMMS-24-09407-T

We would like to thank the referees for their careful review and additional comments on the manuscript. We have made several changes, and our responses are outlined below:

Reviewer #1 (Remarks to the Author):

Authors have made substantial improvements to this manuscript from initial submission, primarily by repeating the airborne transmission experiment and including a second direct contact transmission experiment, including sequencing of nasal wash specimens, and clarifying wording throughout the text. Additional experiments included in this revision investigating the role an adaptation marker in PB2 contributes to virulence and transmission have further resulted in a study which is overall more comprehensive and complete.

Comments:

1) Figure 6: Why did the authors not just report all n=8 wild-type ferrets inoculated with 10^6 TCID50 (panels B and C) together in one graph and compare to the n=4 mutant virus, and not (as it appears) re-copy the mutant virus data across panels B and C both showing results from n=4 from each study?

Response: Thank you for this constructive comment. We have combined the data for ferrets infected with the wild-type virus (n=8) at an inoculation dose of 10^6 TCID50 and compared the results to that for the mutant virus (n=4). The results are the same with significant differences observed in the AUC, but not peak titers for the two viruses. We have revised Figure 6 to reflect these changes.

2) Additionally, for area under the curve analyses, which day span of NW collections were included in this analysis, and how did authors account for animals reaching terminal endpoints prior to full collection of all NW specimens (75% of ferrets inoculated with 10^2 TCID50, and 25% of ferrets inoculated with 10^6 TCID50, were euthanized prior to collection of NW specimens that were negative for influenza virus)? Additional specificity in the methods regarding how this analysis was conducted and how authors controlled for situations where animals were euthanized during sample collection would be helpful (especially as the datapoints for the PB2 A271T area under the curve measurements in panels E and F appear to differ despite what appears to be identical datasets for this virus in panels B and C).

Response: For AUC analyses, we restricted the analyses to days 1-7 post-infection as data was available for all animals at these time points. We have added a section to the methods entitled "Data and Statistical Analyses" that describes how the analysis was performed. Figure 6 was also revised to combine the data for the two studies in which ferrets were inoculated with 10^6 TCID50 of the wild-type virus. The viral titers from this combined dataset were then compared to that for ferrets infected with 10^6 TCID50 of the virus carrying the PB2 A271T virus.

3) Because the serology data is helpful to contextualize the three transmission experiments, and the serology tables are all very small, consider including/embedding Tables 1, 2, and 4 into the same display images as Figures 1, 2, and 5 so the serology data is presented alongside the viral titer information (or combining all serology results into one stand-alone table).

Response: Thank you for feedback on clarifying the presentation of the data. We evaluated adding the serology data to the figures and found the size of the figure panels had to be reduced making it difficult to evaluate the data. Therefore, we simplified the presentation of the serology data by combining it into a single table. This is now Table 1.

4) footnotes in Tables 1 and 2 indicating a lack of expected serological titers for specific animals are better described in the text itself as these assumptions do not reflect actual quantifiable data.

Response: This data is described in the text in lines 137-142, 150, and 241, and we have included it at the bottom of Table 1 as well.

Reviewer #3 (Remarks to the Author):

The authors evaluated the pathogenicity and transmission potential of a clade 2.3.4.4b A(H5N1) virus [A/mink (H5N1)], which caused an outbreak in a mink farm in 2022. The virus was re-constructed by synthesized sequences using reverse-genetics and was tested using the ferret model commonly used for influenza risk assessment studies. The A/mink (H5N1) was transmitted from 3 out of 4 pairs of the inoculated donors to co-housed direct contact (DC) ferrets (75%) and from 3 out of 8 pairs of the inoculated donors to respiratory contact (RC) ferrets (37.5%). Under serial dilution, the ferret infectious doses was determined as 3 TCID₅₀ (based on viral shedding from inoculated animals) and 1 TCID₅₀ (based on clinical signs), which were comparable to that of the A/HK/1/1968 A(H3N2) and A/CA/07/2009 A(H1N1)pdm09 viruses. The A/mink (H5N1) virus possess an adaptive change, PB2-T271A, that has been shown to increase viral polymerase activity in mammalian cells. To evaluate the role of the PB2-T271A change on viral pathogenicity and transmissibility, the authors generated a recombinant virus carrying PB2-271T that is consensus among avian influenza viruses. The A/mink (H5N1) PB2-A271T virus was transmitted to 1 out of 4 RC (25%) and decreased lethality (from 100% to 25%) in ferrets under the inoculation dose of 100 TCID₅₀. Comparable peak viral loads were detected from ferrets inoculated with A/mink (H5N1) and A/mink (H5N1) PB2-A271T; however, the total amount of virus shed (area under the curve) by the A/mink (H5N1) inoculated ferrets were higher. Based on the results, the authors concluded that the clade 2.3.4.4b virus possess heightened pandemic potential relative to ancestral A(H5N1) strains.

Comments:

1) The results have contributed to information to assess the pandemic risk of clade 2.3.4.4b H5N1 viruses, which share genetically similar HA gene but are otherwise genetically diverse in other gene segments. To date, several publications have reported risk assessment study of clade 2.3.4.4b in ferrets. Kendeil et al. (PMID: 37248261) characterized two clade 2.3.4.4b H5N1 viruses isolated from USA and reported no transmission to DC ferrets. Further characterization of the genetically diverse clade 2.3.4.4b viruses observed different in pathogenicity in ferrets. Kobasa et al. (<https://www.researchsquare.com/article/rs-2842567/v1>) have also evaluated the transmission potential of clade 2.3.4.4b H5N1 viruses isolated from Canada, they reported that one virus was able to transmit efficiently to DC but not to RC ferrets. As the H5N1 outbreak in the mink farm has raised concerns for viral transmission potential in mammalian species, Maemura et al. (PMID: 37812908) have similarly constructed 2 recombinant viruses isolated from the same mink farm outbreak and tested their transmission potential in ferrets. They reported no reported no transmission of both viruses to RC ferrets. Taken together, the high genetic diversity of the clade 2.3.4.4b viruses may affect viral transmissibility and pathogenicity in ferrets. As such, the conclusion made by the authors that “clade 2.3.4.4b virus possess heightened pandemic potential relative to ancestral A(H5N1) strains” should be modified based on published literatures.

Response: As suggested by the reviewer, we have modified the conclusions of the manuscript. Line 332-335 previously stated: “In conclusion, this is the first report of both direct contact and limited airborne transmission in a mammalian model of a subclade 2.3.4.4b H5N1 virus, and these findings indicate heightened pandemic potential of this clade of viruses relative to ancestral strains.” This sentence (lines 365-368) now states “In conclusion, this is the first report of both direct contact and limited airborne transmission in a mammalian model of a subclade 2.3.4.4b H5N1 virus indicating these viruses pose a significant pandemic threat.”

2) Please discussed other published risk assessment study of clade 2.3.4.4b viruses in ferrets.

Response: Thank you for this constructive comment. We have added a description of additional risk assessment studies performed on clade 2.3.4.4b viruses in the discussion (lines 296-320).

3) H5N1 viruses have been reported to cause systemic spread in mammalian species and are generally replicating better in the lungs than in the nasal cavity. As such, viral load detected in the nasal washes may not fully reflect viral replication potential in ferrets. To confirm the effect of PB2-T271A change on viral pathogenicity in ferrets, the authors should compare viral titers in the lungs and other organs from ferrets inoculated with A/mink (H5N1) vs. A/mink (H5N1) PB2-A271T.

Response: In the studies presented within the manuscript we did not collect tissue samples or evaluate tissue tropism. Therefore, to address the reviewers' comments, we have stated in lines 353-355 that future studies are warranted to assess viral replication and systemic dissemination of this virus.

4) In regards of transmissibility, A/mink (H5N1) vs. A/mink (H5N1) PB2-A271T showed 25% and 37.5% transmissibility to RC ferrets, which is not statistically significantly different. Can the authors determine the DC transmission potential of A/mink (H5N1) PB2-A271T?

Response: In our studies we did not evaluate direct contact transmission of the PB2 A271T virus. To address the reviewers' comment, we have stated in lines 355-357, "evaluating the role of the PB2 T271A mutation in the context of direct contact transmission will yield additional insight on the contribution of this mutation to the overall transmissibility of the virus."

5) Line 296. The authors mentioned a study that compared the reproducibility of influenza transmission experiments. Can the authors provide a reference to this study?

Response: The appropriate reference was added (Ref 30) to line 323.

6) Figure 6B,C,D,F, since the animals were inoculated with the same dose, would it be possible to combine the results to get a larger animal sample size, which may help to improve statistical analysis?

Response: Reviewer 1 also suggested Fig 6 be revised. Therefore, we combined the data and repeated the analysis. Once this analysis was complete, we revised Fig 6 to reflect these changes.